# Large-capacity and Receiver Authenticable Generative Image Steganography

**Jiannian Wang** [1]  **Yao Lu** [1]  **Guangming Lu** [1]

## Abstract

Diffusion-based generative image steganography converts the input single secret image into noise, and generates the stego image with it serves as the initial noise. Nevertheless, existing methods exhibit three severe limitations: (1) the fixed hiding space constrains their capacity to one secret image; (2) severe inter-secret interference arising from substantial information divergence among multiple secret images while concealing them within a shared hiding space; (3) security risks owing to the absence of the receiver-side verification mechanism. To systematically address these issues, this paper proposes a novel **Receiver Authenticable Generative Image Steganography framework** based on diffusion models. We introduce a **Dynamic Cover Selection and Optimization Engine** to adaptively allocate suitable hiding spaces for different secret images. This design permits the concealment of disparate secret images (or fragments of a single image) into separate spaces, enabling dynamic multi-image concealment while effectively preventing inter-secret interference and expanding capacity through the enlarged hiding spaces. Furthermore, a **Signature and Authentication Controller** cryptographically signs the secret container after concealing and verifies it before extraction, ensuring secure receiver isolation and precise localization of the secret data container. Experiments demonstrate that the proposed framework achieves superior secure multi-receiver isolation and high-performance generative image steganography with large capacity.

[1]Department of Computer Science and Technology, University of Harbin Institute of Technology (Shenzhen), Shenzhen, Guangdong, China. Correspondence to: Yao Lu <luyao2021@hit.edu.cn>, Guangming Lu <luguangm@hit.edu.cn>.

*Proceedings of the 43rd International Conference on Machine Learning*, Seoul, South Korea. PMLR 306, 2026. Copyright 2026 by the author(s).

## 1. Introduction

To mitigate the risks of interception and leakage by malicious third parties during the transmission of private data, image steganography has become a prominent subject in information security research. This technique conceals secret data within images that retain an ordinary appearance, thereby addressing privacy and security concerns associated with the rich visual information such images convey.

Current image steganography methods are primarily divided into **cover-based** and **generative** steganography methods. The former, such as LSBM (Mielikainen, 2006), HUGO (Pevnỳ et al., 2010), UNIWARD (Sameer & Naskar, 2018), and other deep learning based methods (Baluja, 2017; 2019; Lu et al., 2021; Jing et al., 2021; Guan et al., 2022), require external cover image for concealing secret data, exhibiting dependency on cover image. In contrast, generative steganography generates stego images directly from secret data, eliminating the need for a cover image and has attracted more attention. They primarily employ generative models, including flow models, GANs, and diffusion models, to generate the stego images. Owing to the superior image generation quality of the diffusion models than other models, diffusion-based generative image steganography (Yu et al., 2024; Yang et al., 2024; Xu et al., 2025) has become the mainstream approach in generative steganography.

Current diffusion-based generative steganography methods operate by mapping a single secret image to a noise prior, which is then serves as the initial noise of the diffusion model for stego image generation. However, these methods suffer from three severe limitations:

- **Fixed Hiding Space:** These methods solely utilize the initial noise of the diffusion model as a fixed hiding space for secret data, as illustrated in Figure 1. This inflexible and limited hiding space severely restricts the steganography capacity, as the similarity of the latent noise across different images (Figure 2) means the fixed space cannot simultaneously host the distinct noise variables derived from multiple secret images.

- **Multi-Secret Interference:** The concealment of multiple secret images, which exhibit substantial informational differences as shown in Figure 2, within the same fixed space induces competition for the hiding

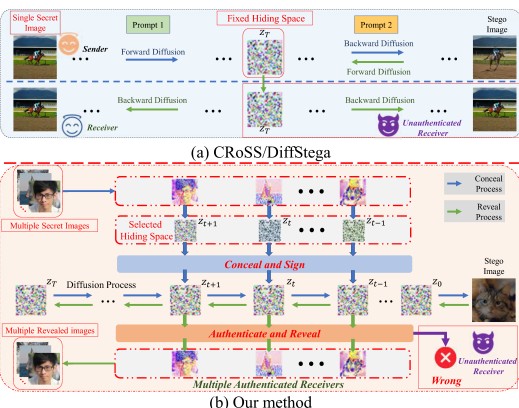

(a) CRoSS/DiffStega

(b) Our method

*Figure 1.* The pipeline comparison of different methods. (a) CRoSS/DiffStega conceals a secret image by first converting it into noise within a fixed hiding space under prompt guidance, which is then used as the initial state to generate a prompt-conditioned stego image. (b) In contrast to existing approaches, the proposed method dynamically selects a unique hiding space for each secret image and enforces receiver isolation through cryptographic authentication. This ensures that each recipient can only locate and reveal the specific image intended for them from the stego image.

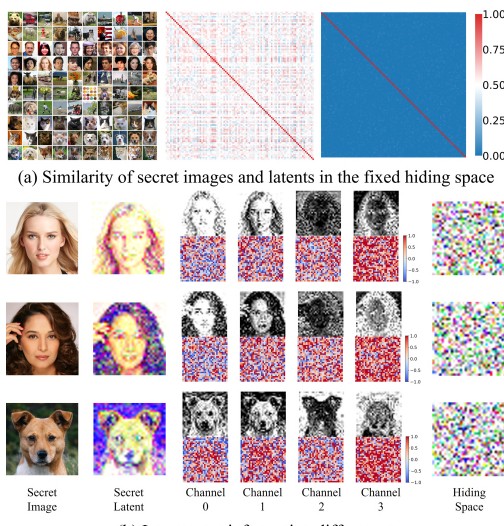

(a) Similarity of secret images and latents in the fixed hiding space

(b) Inter-secret information differences

*Figure 2.* (a) The similarity between different secret images (middle column) and their latents in the fixed space (rightmost column) implies that their corresponding latent variables exhibit considerable divergence. (b) Inter-secret information differences among different secret images. Columns 3–6 illustrate both the per-channel visualizations of the secret latent and their corresponding one-hot maps, while Column 7 visualizes the latent representation of the secret image within the fixed hiding space employed by existing methods. Significant information divergence exists both across different images and different channels within a single image.

space, leading to significant interference and further limiting the expansion of steganography capacity.

- **Security Risk:** Existing methods lack receiver-side authentication and, critically, allow any party with knowledge of the hiding space to extract the concealed data as in Figure 1, thereby exposing a severe security risks.

To systematically address the aforementioned issues, this paper proposes a novel **Receiver Authenticable Generative Image Steganography framework** based on diffusion models, featuring dynamic hiding space selection for multiple secret images as illustrated in Figure 1. In contrast to the static hiding spaces in prior diffusion-based steganography, the proposed method **dynamically selects the optimal hiding spaces for different secret images or distinct fragments of the same secret image**. This capability supports both adaptive secret data hiding and the prevention of inter-secret information interference. This design simultaneously expands the search space for third-party attackers, complicating the efforts of unauthorized parties to locate the concealed data and enhancing overall system security. Furthermore, a **Signature Authentication Controller** is introduced to sign and authenticate the hiding spaces where secret images are hidden. The reveal process is thereby gated and ensures that only authorized receivers can successfully locate and reveal the secret images. This mechanism provides an essential layer of authentication, significantly reinforcing the overall security. Experimental results demonstrate that the proposed model surpasses state-of-the-art methods in steganography performance, capacity, and security. The primary contributions of this work are:

- We propose a novel **Receiver Authenticable Generative Image Steganography framework** based on the diffusion models to support receiver authentication, for the **first time**, within the large-capacity and dynamic secret data generative steganography.

- We propose a **Dynamic Cover Selection and Optimization Engine** to assign the optimal cover for each secret image and prevent the information interference both among multiple secret images and between the secret image and cover.

- We propose a **Signature and Authentication Controller** to achieve separable receiver isolation, ensuring reliable localization and recovery of the concealed secret data only by authenticated receivers and guaranteeing robust security against unauthorized access.

## 2. Related Work

Generative steganography generates stego images directly from secret data using neural networks, thereby eliminating the reliance on carrier images. Previously, the main models used were encoder-decoder networks, GANs (Zhu et al., 2017), and Flow (Wei et al., 2022) models. With the excellent performance of diffusion models in image generation, diffusion models have gradually demonstrated strong

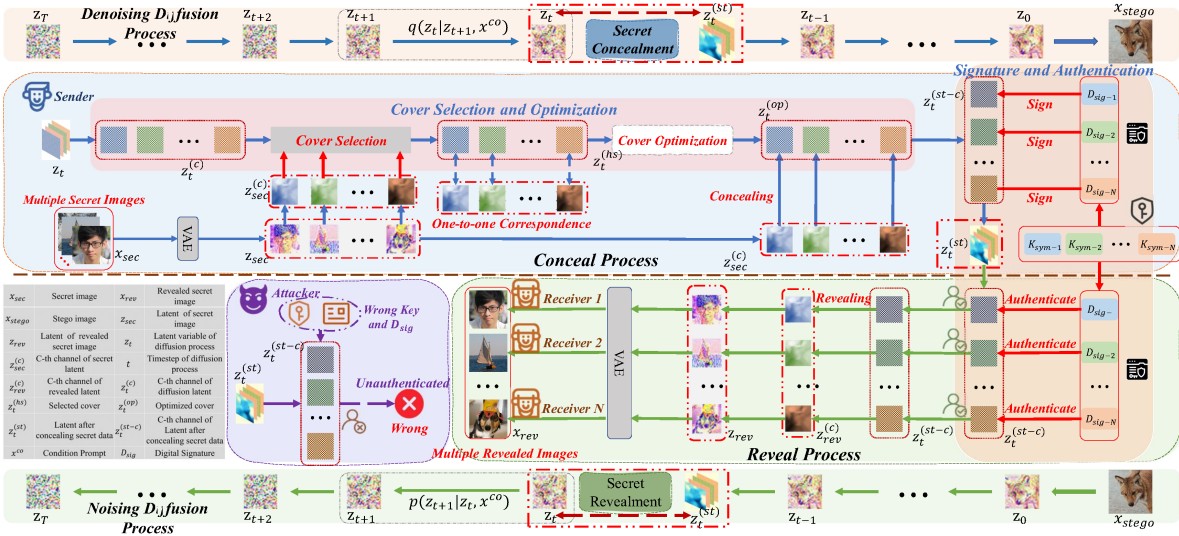

**Figure 3.** The overall structure of the proposed model. While concealing, the secret image $x_{sec}$ is first encoded and decomposed into fine-grained fragments. Joint with the channel selection and cover optimization mechanism, these fragments are then dynamically concealed. Subsequently, the container of the concealed data is signed with a digital signature. The stego image $x_{stego}$ is ultimately generated by completing the remaining diffusion steps. In the reveal stage, only containers that pass the channel-wise authentication are used to accurately reveal the secret image $x_{sec}$.

capabilities in generative steganography tasks. CRoSS (Yu et al., 2024) and DiffStega (Yang et al., 2024) use diffusion models to achieve the steganography of secret images. They use text prompts and image prompts to guide the generation of stego images, enabling stable and controllable generative steganography. Among them, the text prompt is used as a key to provide security for the steganography task. More detailed related work is documented in Appendix A.

Nevertheless, current generative steganography methods are limited by a fixed hiding space that caps capacity and the lack of a receiver-side authentication mechanism. To overcome these limitations, this paper develops a novel receiver authenticable generative image steganography framework that supports the dynamic selection of hiding spaces for both different secret images and fragments of a single image.

## 3. Proposed Methods

### 3.1. Framework

This paper proposes a novel Receiver Authenticable Generative Image Steganography framework, with the pipeline presented in Figure 3. In the conceal process, the input secret image $x_{sec}$ is encoded with the Encoder to get $z_{sec}$, which will be decomposed into fine-grained fragments $\{z_{sec}^{(c)}\}_{c=0}^{C-1}$. For each fragment $z_{sec}^{(c)}$ and a randomly selected timestep $t$, the similarity between $z_{sec}^{(c)}$ and each channel of the intermediate diffusion variable $z_t$ is computed across all channels. The channel with the highest similarity is selected and optimized as the cover for the secret data $z_{sec}^{(c)}$. After concealing,

a digital signature $D_{sig}$, generated with the key, is used to sign the channel containing the secret data. After all fragments are concealed, the diffusion process completes the generation of $x_{stego}$ with Gaussian noise $z_T \sim \mathcal{N}(0, \mathbf{I})$ as the initial input. In the reveal phase, the stego image $x_{stego}$ is encoded and fed into the diffusion model. The resulting latent variables $\{z_t\}_{t=0}^T$ are then channel-wise authenticated with the same signature $D_{sig}$. Only authenticated channels are utilized to reconstruct the revealed secret image $x_{rev}$.

### 3.2. Generative Image Steganography with Adaptive Hiding Spaces

Existing steganography models confine the concealment of secret data to the latent variables of a fixed timestep; for instance, CRoSS and DiffStega conceal data solely within the initial noise. This static hiding space poses two issues: security, as it offers a static target for attackers, and compatibility, as forcing diverse data into one space causes conflict.

To simultaneously overcome these limitations, the proposed framework employs a dynamic hiding strategy. It dynamically selects distinct hiding spaces within the generative process, each tailored to a specific secret data. This obscures the location of concealed data and minimizes the information conflicts inherent in conceal multiple images into a fixed space. The formal definition for **Generative Image Steganography with Adaptive Hiding Spaces** is:

***Problem Definition.*** Let $\mathcal{S} = \{S_1, S_2, \cdots, S_M\}$ denote a set of $M$ secret images need to be concealed, each of size $N_s \times N_s$. $\mathcal{Z} = \{Z_1, Z_2, \cdots, Z_N\}$ represents the set of $N$

spaces that can be used to hide secret data, satisfying $M \leq N$. Let $\mathcal{K} = \{K_1, K_2, \cdots, K_M\}$ be the set of keys shared between the sender and the receiver. The steganography objective of this problem is to generate the stego image $x_{stego}$ with the model $\mathcal{G}$, based on the secret images $\mathcal{S}$ and keys $\mathcal{K}$, by dynamically concealing the secrets within the space set $\mathcal{Z}$. This process is formulated as

$$x_{stego} = \mathcal{G}(\mathcal{S}, \mathcal{Z}, \mathcal{K}),$$

which must be invertible to reveal the secret images $\mathcal{S}$ as

$$\mathcal{S} = \mathcal{G}^{-1}(x_{stego}, \mathcal{Z}, \mathcal{K}),$$

where $\mathcal{G}^{-1}$ is the inverse process of the generate process.

Specifically, $\mathcal{G}$ indicate the diffusion model in this paper. The hiding space $\mathcal{Z}$ is the collection of intermediate latent variables across all diffusion timesteps and channels, *i.e.*, $\mathcal{Z} = \{\{z_t^{(i)}\}\}, t = 0, 1, \cdots, T, i = 0, 1, \cdots, C$, where $C$ is the number of channels in the latent variables $z_t$. This section illustrates the conceal process and reveal process, which are presented in Algorithm 1 and Algorithm 2, respectively.

### 3.2.1. CONCEAL PROCESS

The conceal process is initiated with a random standard Gaussian noise $z_T \sim \mathcal{N}(0, \mathbf{I})$. Diffusion models employ a denoising trajectory to incrementally transform this noise into structured data distributions through sequential latent variable refinement with learnable transition probabilities $q(z_{t-1}|z_t, x^{co})$. With the EDICT framework, the intermediate variables in the denoising trajectory can be obtained through stepwise iteration of the denoising process in the diffusion model guided by prompts $x^{co}$:

$$
\begin{cases}
z_t^{in} &= a_{t+1} \cdot z_{t+1} + b_{t+1} \cdot \epsilon(z_{t+1}, t+1, x^{co}), \\
y_t^{in} &= a_{t+1} \cdot y_{t+1} + b_{t+1} \cdot \epsilon(z_t^{in}, t+1, x^{co}), \\
z_t &= p \cdot z_t^{in} + (1-p) \cdot y_t^{in}, \\
y_t &= p \cdot y_t^{in} + (1-p) \cdot z_t,
\end{cases}
\tag{1}
$$

where $t \in \{0, 1, \cdots, T-1\}$ is the timestep of the diffusion process and $\epsilon$ is the diffusion model.

While concealing within timestep $\tilde{t} \in \tau$, the secret image $x_{sec}$ is mapped by an encoder $\mathbf{E}$ to a latent representation with dimensionality $C$

$$z_{sec} = \mathbf{E}(x_{sec}) \tag{2}$$

firstly for dimensionality reduction of the secret image, where $\tau \subset \{0, 1, \cdots, T-1\}$ is the timesteps for concealing.

To achieve a more fine-grained conceal, the latent representation $z_{sec}$ is decomposed into $C$ fragments $z_{sec}^{(c)}, c \in 0, 1, \cdots, C-1$ along the channel dimension. Each fragment is then concealed into a selected intermediate variable

---

**Algorithm 1** The Conceal Process

**Input:** The secret image $x_{sec}$, the encoder $\mathbf{E}$, the decoder $\mathbf{D}$, the conceal process $\mathbb{M}$, EDICT framework's conditioned denoising process $\epsilon_\theta$, the prompt $x^{co}$, the number $T$ of time steps for sampling, the timesteps $\tau$ for concealing image, an initial random noise $z_T$, the symmetric key $k_{sym}$, digital signature network $DS$, watermark network $\mathcal{W}$, cover optimize module $\mathcal{O}$.

**Output:**
$z_{sec} = \mathbf{E}(x_{sec})$;
$C \leftarrow the\ channel\ dimension\ of\ z_{sec}$;
$c \leftarrow 0$;
$z_T \sim \mathcal{N}(0, 1)$;
$D_{sig} \leftarrow DS(k_{sym})$; # Digital Signature generate
**for** $t = T-1, \cdots, 2, 1, 0$ **do**
  $z_t \leftarrow \epsilon_\theta(z_{t+1}, t+1, x^{co})$;
  **if** $\tilde{t} == t, \forall \tilde{t} \in \tau$ **then**
    **if** $c < C$ **then**
      $Sim_i \leftarrow Sim(z_{sec}^{(c)}, z_t^{(i)}), i \in \{0, 1, \cdots, C-1\}$;
      $z_t^{hs} \leftarrow z_t^i$ with max $Sim_i$; # Cover Select
      $z_t^{(c)} \leftarrow \mathcal{O}(z_t^{hs})$; # Cover Optimize
      $\tilde{z}_{\tilde{t}}^{(hs)} \leftarrow z_{sec}^{(c)} + \mathbb{M}(z_t^{(c)}, k_{sym}, x^{co})$; # Conceal
      $z_{\tilde{t}}^{(hs)} \leftarrow \tilde{z}_{\tilde{t}}^{(hs)} + \mathcal{W}(\tilde{z}_{\tilde{t}}^{(hs)}, D_{Sig})$ # Sign
      $z_t^{(c)} \leftarrow z_{\tilde{t}}^{(hs)}$;
      $c \leftarrow c + 1$;
    **end if**
  **end if**
**end for**
$z_{stego} \leftarrow z_0$;
$x_{stego} \leftarrow \mathbf{D}(z_{stego})$;

---

along the diffusion trajectory $z_t^{(hs)}$ as

$$
\begin{cases}
z_{\tilde{t}}^{(c)} &= \mathcal{O}(z_{\tilde{t}}^{(hs)}), \\
\tilde{z}_{\tilde{t}}^{(hs)} &= z_{sec}^{(c)} + \mathbb{M}(z_{\tilde{t}}^{(c)}, k_{sym}, x^{co}),
\end{cases}
\tag{3}
$$

where $\mathcal{O}$ is the cover optimization process and $\mathbb{M}$ is the conceal network. This design not only mitigates information conflict between the secret data and the cover, but also multiplies the number of containers containing secret data. In this case, an adversary is forced to analyze all potential covers concurrently, which exponentially increases the analytical complexity and enhances the overall security.

After concealing the single-channel latent representation $z_{sec}^{(c)}$, the corresponding original noise variable $z_t^{(c)}$ in the stego generation process is systematically replaced with tailed secret container $\tilde{z}_{\tilde{t}}^{(hs)}$, achieving channel-specific concealment. This process is iteratively applied to all channels of $z_{sec}$ sequentially until the entire latent is concealed.

To ensure reliable extraction by the receiver, the hidden location must be signed with digital signature $D_{sig}$. This

paper addresses this through a Signature and Authentication Controller (Section 3.4).

The conceal pipeline sequentially processes each secret image in multi-image scenarios, concealing their latent representation into unique timesteps or sub-bands to prevent inter-secret interference. Once all the secret images are concealed, the remaining diffusion steps are iterated to produce the final latent representation $z_{stego}$ of the stego image $x_{stego}$. The stego image is then obtained through decoding the latent with decoder $\mathbf{D}$, formulated as

$$x_{stego} = \mathbf{D}(z_{stego}). \tag{4}$$

By dynamically concealing secret data across the latent structure of the diffusion trajectory, the proposed framework leverages the high-quality generation capability of diffusion models, significantly increase its effectiveness.

Through extensive experimental analysis, the default time-step set $\tau = [0, 1, 2, 3]$ was employed for single-image steganography. In multi-image steganography, sequential time-step set $\tau$ were allocated at four-step intervals starting from $t = 0$; $i.e.$, $\tau_1 = [0, 1, 2, 3]$ and $\tau_2 = [4, 5, 6, 7]$.

### 3.2.2. Reveal Process

In the reveal process, the receiver encodes the stego image $x_{stego}$ into latent representation $z_{stego}$ with encoder $\mathcal{E}$ firstly, formulated as

$$z_{stego} = \mathbf{E}(x_{stego}). \tag{5}$$

These latent variables are then fed into the EDICT model's inverse diffusion process

$$\begin{cases} y_t^{in} &= (y_t - (1-p) \cdot z_t)/p, \\ z_t^{in} &= (z_t - (1-p) \cdot y_t^{in})/p, \\ y_{t+1} &= (y_t^{in} - b_{t+1} \cdot \epsilon(z_t^{in}, t+1, x^{co}))/a_{t+1}, \\ z_{t+1} &= (z_t^{in} - b_{t+1} \cdot \epsilon(y_t, t+1, x^{co}))/a_{t+1}. \end{cases} \tag{6}$$

This process deterministically inverts the forward diffusion trajectory, iteratively recovering the intermediate latent representations with probabilities $p(z_{t+1}|z_t, x^{co})$ at each timestep. The deterministic nature of this process, inherent to the EDICT framework, ensures minimal reveal error.

Channel-wise authentication is conducted at each timestep using the same digital signature $D_{sig}$ (Section 3.4). The receiver verifies the consistency between the hosted signature and the one embedded within each channel. Upon successful authentication, the receiver extracts the concealed fragment $z_{rev}^{(c)}$ from the authenticated channel

$$\begin{cases} \tilde{z}_t^{(c)} &= \mathcal{O}(z_t^c), \\ z_{rev}^{(c)} &= z_t^{(c)} - \mathbb{M}(\tilde{z}_t^{(c)}, k_{sym}, x^{co}). \end{cases} \tag{7}$$

This authentication-extraction cycle iterates across all timesteps and channels until the complete secret fragments $z_{rev}^{(c)}, c \in 0, 1, \cdots, C-1$ are recovered.

---

**Algorithm 2** The Reveal Process

**Input:** The stego image $x_{stego}$, the encoder $\mathbf{E}$, the decoder $\mathbf{D}$, the reveal process $\mathbb{M}$, EDICT framework's noising inversion process $\epsilon_\theta^{-1}$, the prompt $x^{co}$, the number $T$ of time steps for sampling, the symmetric key $k_{sym}$, digital signature network $DS$, Authentication network $\mathcal{V}$, cover optimize module $\mathcal{O}$.

**Output:**

$z_{stego} = \mathbf{E}(x_{stego})$;
$B, C, H, W \leftarrow the\ shape\ of\ z_{stego}$;
$c \leftarrow C - 1$;
$z_{rev} \leftarrow [0]^{B \times C \times H \times W}$;
$D_{sig} \leftarrow DS(k_{sym})$; # Digital Signature generate
$d \leftarrow C - 1$;
**for** $t = 1, 2, \cdots, T$ **do**
    $z_t \leftarrow \epsilon_\theta^{-1}(z_{t-1}, t-1, x^{co})$;
    **for** $c \geq 0$ **do**
        $\tilde{D}_{Sig} \leftarrow \mathcal{V}(\tilde{z}_t^i)$; # Digital Signature Extraction
        **if** $\tilde{D}_{Sig} == D_{sig}$ **then**
            $\tilde{z}_t^c \leftarrow \mathcal{O}(z_t^c)$;
            $z_{rev}^{(d)} \leftarrow z_t^{(c)} - \mathbb{M}(\tilde{z}_t^{(c)}, k_{sym}, x^{co})$; # Reveal
            $z_{rev}[:, d, :, :] \leftarrow z_{rev}^{(d)}$;
            $d \leftarrow d - 1$;
        **else**
            $c \leftarrow c - 1$;
        **end if**
    **end for**
    $c \leftarrow C$;
**end for**
$x_{rev} = \mathbf{D}(z_{rev})$;

---

The extracted fragments are aggregated to reconstruct the latent representation $z_{rev}$, which is then fed into a decoder $\mathbf{D}$ to reconstruct the revealed secret image

$$x_{rev} = \mathbf{D}(z_{rev}). \tag{8}$$

The extraction pipeline operates iteratively for multiple secret images, with each iteration dedicated to recovering a specific fragment authenticated by its unique digital signature. This design ensures that access is cryptographically restricted: a receiver can only extract those secret images for which they hold the corresponding digital signature, thereby preventing the leakage of unauthenticated secret data.

### 3.3. Dynamic Cover Selection and Optimization Engine

Since different channels of the cover latent variables carry distinct information, directly conceal secret data can cause interference. To address this, this work strategically select the most suitable channel for hiding. This involves computing the similarity between the secret data and each channel

$$Sim_i = Sim(z_{sec}^{(c)}, z_t^{(i)}), i \in \{0, 1, \cdots, C-1\}. \tag{9}$$

The channel achieving the highest similarity score, termed as $z_t^{hs}$, is then utilized as the container for concealment, thereby reducing the inherent information conflict between the secret and the cover. In this work, the cosine similarity is utilized as the similarity metric.

Although cover selection mitigates the initial discrepancy and interference between secret data and the cover, it cannot fully eliminate these effects. To address this issue, the proposed framework further optimizes the selected cover by

$$z_t^{hs} = \mathcal{O}(z_t^{hs}), \tag{10}$$

where $\mathcal{O}$ is a DenseNet with 5 Convolution layers. The refined cover is then employed in both the conceal and reveal processes to hide and extract the secret data, respectively.

### 3.4. Signature and Authentication Controller

To enforce receiver isolation and ensure that each receiver can only recover a specific secret image from the stego image, a novel Signature and Authentication Controller is incorporated, drawing inspiration from analogous protocols in secure communications (e.g., TLS). This controller not only achieves strict receiver isolation, preventing any single receiver from accessing all concealed images, but also substantially enhances the overall security of the framework and the concealed secret data. Only authenticated receivers are able to correctly reveal their intended secret images. The overall pipeline of the controller is illustrated in Figure 3.

Specifically, on the **sender side**, after concealing the secret data, the selected hiding space must be signed for reliable localization and authentication by the receiver. Thus, employing a symmetric key $k_{sym}$ and a lightweight convolution network $DS$ that are both pre-shared with the receiver, the sender generates a unique digital signature $D_{Sig}$:

$$D_{Sig} = DS(k_{sym}), k_{sym} \in \mathcal{K}, \tag{11}$$

and conceals it alongside the secret data within the same space, using the watermark network $\mathcal{W}$

$$\tilde{z}_t^i = z_t^i + \mathcal{W}(z_t^i, D_{Sig}). \tag{12}$$

On the **receiver side**, the Authentication network $\mathcal{V}$ is used to extract the concealed digital signature from its container

$$\tilde{D}_{Sig} = \mathcal{V}(\tilde{z}_t^i). \tag{13}$$

The extracted signature is then compared against a reference signature generated locally using the same symmetric key $k_{sym}$ and model $DS$. A successful match ($\tilde{D}_{Sig} = D_{Sig}$) signifies the presence of concealed information within that space, and the receiver could reveal the concealed secret data within it. If not match, the authentication process iterates to the next candidate space until all possibilities are verified. In

this work, the symmetric key generation method in (Wang et al., 2025) is utilized as the default mechanism.

This design ensures that authentication and subsequent revealing of the concealed data are strictly contingent upon possession of the correct symmetric key, thereby restricting access to authorized receivers only. It effectively integrates cryptographic authentication into the dynamic steganography strategy, forming a unified security framework.

### 3.5. Loss Function

The loss function is describe in detail below.

**Conceal loss.** The steganography process intends to generate the stego image $x_{stego}$ based on the secret image $x_{sec}$. For security purposes, the latents of the concealed secret latents at timestep $t$ $\tilde{z}_{\tilde{t}}$ should closely match that of the latents $z_t$ of the diffusion process at timestep $t$. To enforce these constraints, the conceal loss is defined as follows:

$$\mathcal{L}_C = l_s(z_t, \tilde{z}_{\tilde{t}}), \tag{14}$$

where $l_s$ represents the $l_1$ or $l_2$ norm, serving as a measure of the difference between two latents. In this paper, the $l_1$ norm is employed.

**Reveal loss.** To ensure that the revealed latents $z_{rev}$ aligns with the secret image latents $z_{sec}$, the reveal loss is introduced and formulated as:

$$\mathcal{L}_R = l_s(z_{rev}, z_{sec}). \tag{15}$$

**Sign and authentication loss.** To ensure both container invariance after signing and signature consistency during authentication, the sign and authentication loss is introduced

$$\mathcal{L}_{SV} = l_s(z_t^i, \tilde{z}_t^i) + l_s(D_{sig}, \tilde{D}_{sig}). \tag{16}$$

**Total loss.** The total loss function $\mathcal{L}_{Total}$ is defined as the weighted sum of the conceal loss $\mathcal{L}_C$, reveal loss $\mathcal{L}_R$ and Sign and Authentication loss, formulated as:

$$\mathcal{L}_{Total} = \lambda_1 \mathcal{L}_C + \lambda_2 \mathcal{L}_R + \lambda_3 \mathcal{L}_{SV}, \tag{17}$$

where $\lambda_1$, $\lambda_2$ and $\lambda_3$ are trade-off parameters set to 1.0, 1.0, and 0.5, respectively, to balance the different losses.

## 4. Experiments

### 4.1. Experimental Setting

The proposed framework utilizes the pretrained Stable Diffusion V1.5 to generate the stego images. It is implemented with PyTorch and trained on the DIV2K (Agustsson & Timofte, 2017) training dataset. The evaluation is performed on the UniStega (Yang et al., 2024) dataset, DIV2K (Agustsson & Timofte, 2017) test dataset and Stega260 dataset (Yu et al., 2024) at a resolution of $256 \times 256$. More implementation details are presented in the Appendix C.

*Table 1.* Numerical comparisons with generative steganography methods on the UniStega dataset.

| Method | Quantity | Nvidia RTX 4090 | | Nvidia RTX 3090 | | Stego images | | Authenticated receiver | | Unuthenticated receiver | |
|---|---|---|---|---|---|---|---|---|---|---|---|
| | | Time(s) | Memory(GB) | Time(s) | Memory(GB) | PSNR↓ | SSIM↓ | PSNR↑ | SSIM↑ | PSNR↓ | SSIM↓ |
| CRoSS | 1 | 14.41 | 7.43 | 39.38 | 8.58 | 19.03 | 0.66 | 21.25 | 0.71 | - | - |
| DiffStega | 1 | 20.90 | 8.35 | 41.00 | 9.02 | **18.61** | **0.59** | 23.29 | 0.77 | **17.53** | **0.54** |
| DiffStega‡ | 1 | 20.90 | 8.35 | 41.00 | 9.02 | 19.73 | 0.65 | **23.92** | **0.79** | 20.68 | 0.70 |
| **Ours** | 1 | 15.61 | 7.39 | 28.94 | 8.62 | **9.48(9.13↓)** | **0.21(0.38↓)** | **26.59(2.67↑)** | **0.80(0.01↑)** | **9.11(8.42↓)** | **0.23(0.31↓)** |
| | 2 | - | - | - | - | 9.48 | 0.24 | 24.08 | 0.71 | 9.05 | 0.20 |
| | 4 | - | - | - | - | 9.47 | 0.24 | 22.61 | 0.68 | 8.91 | 0.17 |
| | 8 | - | - | - | - | 9.43 | 0.23 | 19.83 | 0.60 | 8.76 | 0.14 |

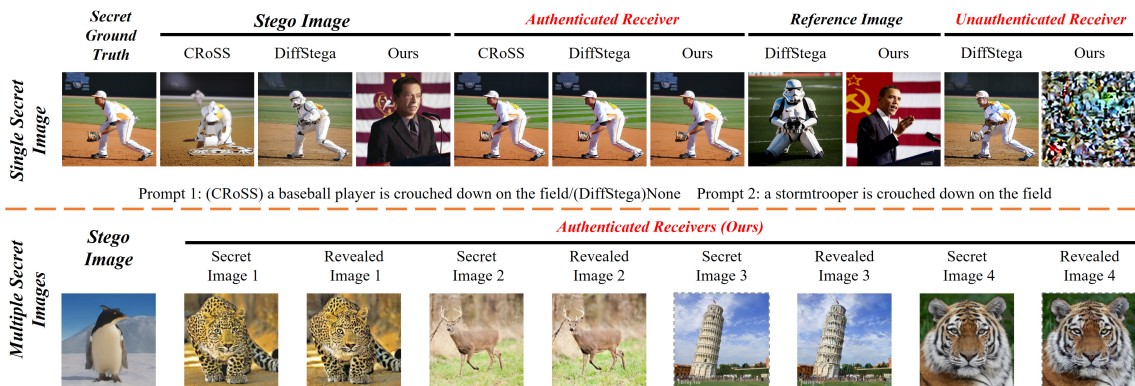

*Figure 4.* Visualization comparison results of the proposed model versus state-of-the-art methods, for both single (top row) and multiple secret images (bottom row). These prompts are utilized in CRoSS and DiffStega, whereas our model functions without text prompts. The reference image is used as the image condition in both DiffStega and our model. The secret images are revealed with authenticated and unauthenticated receivers.

## 4.2. Single Image Steganography

The proposed method is evaluated against other generative image steganography models, with numerical results displayed in Table 1 and Table 2. Table 1 presents the quantitative results in comparison with other methods on the UniStega dataset and Table 2 is the results on the DIV2K and Stega260 dataset. Specifically, for authenticated receivers, the proposed model achieves an impressive PSNR gain of **2.67** dB for recovery/secret pairs with a marginal impact on SSIM, indicating a substantial enhancement. Conversely, for secret/stego image pairs, it reduces PSNR by **9.13** dB and SSIM by **38%** compared to other methods, achieving a significant reduction of the similarity between stego and their corresponding secret images. Evidently, the proposed method excels in conceal and reveal process compared to other generative steganography models, delivering high-quality revealed secret images. More visualization results for single image steganography are shown in Appendix D.

## 4.3. Multiple image Steganography

**Quantitative results**. To evaluate multi-image steganography efficacy, the proposed framework is evaluated in the multi-image scenario. The experimental results in Table 1 and Table 2 demonstrate the effectiveness of the proposed method for large-capacity steganography. Crucially, the proposed framework conceals multiple secrets without com-

*Table 2.* Numerical results on the DIV2K and Stega260 dataset.

| Dataset | Quantity | Stego Image | | Revealed Image | |
|---|---|---|---|---|---|
| | | PSNR↓ | SSIM↓ | PSNR↑ | SSIM↑ |
| DIV2K | 1 | 9.71 | 0.15 | 22.34 | 0.71 |
| | 2 | 9.70 | 0.15 | 21.03 | 0.64 |
| | 4 | 9.69 | 0.15 | 19.47 | 0.59 |
| | 8 | 9.67 | 0.15 | 18.02 | 0.52 |
| Stega260 | 1 | 9.56 | 0.29 | 26.87 | 0.81 |
| | 2 | 9.55 | 0.29 | 24.31 | 0.76 |
| | 4 | 9.55 | 0.29 | 22.73 | 0.68 |
| | 8 | 9.52 | 0.28 | 20.17 | 0.61 |

promising performance or security. For instance, with two secret images, its reconstruction quality (**24.08 dB**) matches that of DiffStega with one image (**23.92 dB**); the proposed method obtains **22.61 dB** while conceal foue images, which is on par with CRoSS (**21.25 dB**) with one secret image. These findings demonstrate the capability of the proposed framework for large-capacity steganography.

**Qualitative Results.** Figure 4 provides a visual comparison for single-image (top row) and multi-image (bottom row) steganography of the proposed model and other generative steganography models. While single-image results are evaluated on UniStega, multi-image performance is assessed on Stega260. The bottom row of the figure shows that the proposed method successfully conceals multiple secret images within a single stego image through generative steganog-

*Table 3.* NIQE scores for various models.

| | ORIGINAL IMAGE | ISN | HINET | CROSS | DIFFSTEGA | **OURS** |
|---|---|---|---|---|---|---|
| NIQE↓ | 5.14 | 11.28 | **5.35** | 5.60 | 5.58 | **5.49** |

*Table 4.* The detection accuracy (%) of various methods as evaluated by SRNet, XuNet, and YeNet.

| | COVER-BASED METHOD | | | | GENERATIVE METHOD | | |
|---|---|---|---|---|---|---|---|
| | WENG | UDH | ISN | HINET | CROSS | DIFFSTEGA | OURS |
| SRNET | 81.47 | 76.34 | 54.86 | 53.52 | 51.73 | 51.68 | **50.57(1.11↓)** |
| XUNET | 85.31 | 79.82 | 55.68 | 55.54 | 52.10 | 52.03 | **50.86(1.17↓)** |
| YENET | 86.24 | 80.26 | 55.42 | 55.37 | 53.12 | 52.86 | **50.71(2.15↓)** |

raphy. In contrast, CRoSS and DiffStega are limited to single-secret concealment. These visual results highlight the efficacy of the proposed method for multi-image steganography. Collectively, these results validate that the proposed Dynamic Cover Selection and Optimization Engine and the Signature and Authentication Controller systematically constitute a complementary system and jointly deliver both large-capacity steganography and secure receiver isolation. **More Visualization results for multi-image steganography are illustrated in Appendix D.**

### 4.4. Security Analysis

**Naturalness and Imperceptibility.** The Natural Image Quality Evaluator (NIQE) quantifies visual naturalness and security without the assistance reference images or human feedback. The proposed method achieves a 0.14 lower NIQE score than other generative steganography models. This indicates that the proposed model produces stego images with superior naturalness and imperceptibility compared to other generative steganography models.

**Steganography analysis.** The anti-steganalysis ability is a critical security metric for image steganography, measuring the difficulty of distinguishing stego images from covers using steganalysis tools. To assess this capability, the evaluation is conducted utilizing the steganalysis tool StegExpose (Boehm, 2014) and three steganalysis networks: SRNet(Boroumand et al., 2018), XuNet (Xu et al., 2016), and YeNet (Ye et al., 2017). The evaluation results are presented in Figure 5 and Table 4 respectively. Lower detection accuracy and a smaller area under the curve (AUC) correspond to better security performance. These results indicate that the proposed model achieves superior anti-steganalysis performance than compared steganography methods, demonstrating its enhanced security. **Further security analysis is provided in Section B.**

### 4.5. Ablation Study

**Effect of the Cover Selection.** To match secret data with optimal covers, a cover selection mechanism is incorporated into the proposed framework. As evidenced by Table 5, this

*Table 5.* Effectiveness of Cover Selection and Cover Optimization.

| CHANNEL SELECTION | COVER OPTIMIZATION | PSNR | SSIM |
|---|---|---|---|
| ✗ | ✗ | 20.19 | 0.54 |
| ✓ | ✗ | 23.87 | 0.71 |
| ✗ | ✓ | 23.41 | 0.71 |
| ✓ | ✓ | **26.59** | **0.80** |

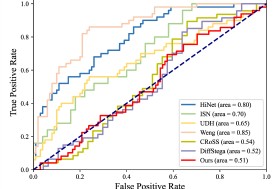 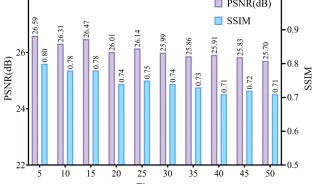

*Figure 5.* Security performance detected by StegExpose.

*Figure 6.* Ablation on the selected timestep for concealing.

mechanism significantly enhances the revealed secret image PSNR from 20.19dB to 23.87dB. Collectively, these results validate that the mechanism is essential to improving the performance of generative image steganography.

To further validate the effectiveness of the proposed cover selection, the t-SNE clustering experiment was conducted, with the visualization results in Figure 7. The figure visualizes the latent variable distributions before and after concealing different secret images using various methods within a fixed hiding space. The failure of CRoSS and DiffStega to resolve inter-secret conflicts is evidenced by their divergent clusters, underscoring the necessity for secret-specific cover selection. In contrast, the proposed framework induces a clear convergence of the clusters. This outcome validates its efficacy in mitigating inter-secret information interference.

**Impact of the selected hidden timesteps.** Extensive experiment investigates the impact of concealing secret images at varying time steps (Figure 6), measuring steganography efficacy through the reconstruction quality (PSNR/SSIM) of secret images. As shown in Figure 6, the performance degrades progressively with increased hiding time steps, due to error accumulation in the inverse diffusion process. Notably, the proposed method retains a performance advantage over comparative methods across all time steps, including those maximizing error. These results empirically demonstrate the superior steganography capability of the proposed methodology. Furthermore, the proposed framework exhibits enhanced robustness as indicated with **more Ablation Study** presented in Appendix D.

**Impact of the selected diffusion backbones.** To evaluate generalization capability, experiments were performed using SD2.1-base at 512×512 resolution. Detailed results are provided in Table 6. For the single-image steganography task, the proposed method achieves superior performance over DiffStega and CRoSS, improving PSNR from 23.87dB to 26.23dB and increasing SSIM from 0.78 to 0.81. These

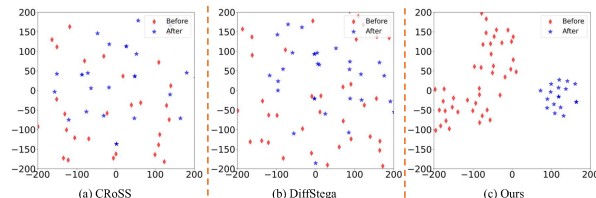

*Figure 7.* Comparison of t-SNE clustering visualizations for latent variables before and after steganography with different methods.

*Table 6.* Generalization across Diffusion Architectures and Image Resolutions.

| MODEL | RESOLUTION | PSNR | SSIM |
|---|---|---|---|
| CROSS | $512 \times 512$ | 21.31 | 0.70 |
| DIFFSTEGA | $512 \times 512$ | 23.87 | 0.78 |
| OURS | $512 \times 512$ | 26.23 | 0.81 |

results demonstrate that the core functional modules, the dynamic cover selection and optimization engine, along with the signature and authentication controller, are engineered to be independent of the diffusion backbone. This decoupling ensures seamless transferability to alternative diffusion architectures (e.g., SD2.0, SDXL, DiT) and higher resolutions (e.g., $512 \times 512$).

## 5. Conclusion

This paper presents a Receiver Authenticable Generative Image Steganography framework based on the diffusion model, to circumvent the limitations of existing generative image steganography methods. The proposed model introduces the Dynamic Cover Selection and Optimization Engine for dynamically selecting hiding spaces for multiple secret images and reduces the information conflict between the secret image and cover. Furthermore, to guarantee that the receiver can accurately locate and reveal the secret data, the Signature and Authentication Controller is proposed to mark the secret data container during data concealment and to verify this sign while revealing. Extensive experiments demonstrate that the proposed framework significantly outperforms SOTA generative steganography methods in both effectiveness and security.

## Acknowledgements

This work was supported in part by the NSFC fund (NO. 62572145), in part by the Shenzhen Key Technical Project (NO. ZDCY20250901111404005, JCYJ20241202123728037, NO. KJZD20240903100712017), in part by the Guangdong Natural Science Fund Project (NO. 2026A1515011491), in part by the Shenzhen Fundamental Research Fund (NO. JCYJ20210324132210025), and in part by the Natural Science Foundation of Shenzhen General Project under Grant JCYJ20240813110007010.

## Impact Statement

This paper delves into an exploration of large-capacity receiver authenticable generative image steganography technology, with the objective of generating the stego images from secret images. Through a succession of innovative algorithmic designs and meticulous experimental analyses, it has significantly boosted the security and effectiveness of generative image steganography methods. The integration of the receiver authentication has effectively forged an initial connection between image steganography and cryptology. This linkage not only enriches the theoretical framework of image steganography but also endows cryptology with new application scenarios.

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

# A. Detailed Related Work

## A.1. Image Steganography

**Cover-based Steganography.** Conventional cover-based steganographic techniques, including Least Significant Bit (LSB) (Mielikainen, 2006) substitution, operate by embedding secret data within either the spatial or transform domains of cover media. The advent of deep learning has catalyzed the development of deep learning steganography, which exhibit superior embedding efficiency and significantly enhanced payload capacity. The pioneering work of Baluja (Baluja, 2017; 2019) introduced DDH, the first encoder-decoder architecture capable of full-image conceal, while UDH (Zhang et al., 2020) subsequently proposed an alternative neural conceal paradigm. Subsequent advancements in Invertible Neural Networks enabled ISN (Lu et al., 2021) to establish new benchmarks in high-capacity steganography. Further extensions (Jing et al., 2021; Guan et al., 2022; Zhou et al., 2025) explored multi-image steganography through reversible architectures, achieving additional capacity gains.

Nevertheless, the intrinsic reliance on cover image inherent introduces operational limitations in practical application. In contrast, this paper establishes a novel secure multi-image generative steganography method that obviates the requirement for external cover images.

**Generative Steganography.** Generative steganography generates stego images directly from secret information using specific neural networks, thereby eliminating the reliance on carrier images. Previously, the main models used were encoder-decoder networks, GANs (Zhu et al., 2017), and Flow (Wei et al., 2022) models. With the excellent performance of diffusion models in image generation, diffusion models have gradually demonstrated strong capabilities in generative steganography tasks. GSD (Wei et al., 2023) uses a diffusion model to generate stego images from binary secret information, showing excellent steganography capabilities. On the other hand, CRoSS (Yu et al., 2024) and DiffStega (Yang et al., 2024) use diffusion models to achieve the steganography of secret images. They use text prompts and image prompts to guide the generation of stego images, enabling stable and controllable generative steganography. Among them, the text prompt is used as a key to provide security for the steganography task.

Nevertheless, current generative steganography methods are limited by a fixed hiding space that caps capacity and the lack of a receiver-side authentication mechanism. To overcome these limitations, this paper develop a novel Receiver Authenticable Generative Image Steganography method that supports the dynamic selection of hiding spaces for both different secret images and different fragments of a single image.

## A.2. Diffusion Model

Diffusion models (Ho et al., 2020; Song et al., 2021) have emerged as a prominent class of generative architectures, demonstrating substantial advancements in image generation. Recent innovations (Chahine & Kim, 2024; Oguz et al., 2024; He et al., 2024; Bartosh et al., 2024; Nobis et al., 2024; Motwani et al., 2024) have further expanded the capabilities of diffusion models. The ControlNet (Zhang et al., 2023) architecture implements modular adapter components to enable conditional generation through semantic segmentation maps, skeletal pose estimations, and edge detection constraints. In addition, the IP-Adapter (Ye et al., 2023) framework incorporates reference images as conditional inputs, utilizing learned image embeddings to guide the generative process. To mitigate inversion inaccuracies, Null-text Inversion (Hertz et al., 2023) employs auxiliary fine-tuning procedures to optimize null-space embeddings, while EDICT (Wallace et al., 2023) implements coupled transformation mechanisms to achieve precise diffusion inversion without parameter optimization. The BDIA (Zhang et al., 2024) sampler employs a symmetric bidirectional integration architecture to ensure precise inversion fidelity. BELM (Wang et al., 2024) proposes a generalized theoretical framework for bidirectional explicit linear multi-step sampling to achieve exact diffusion inversion.

This paper presents a novel generative image steganography framework based on diffusion models. The proposed method not only enables the dynamic selection of hiding spaces for multiple secret images and the fragments of a single secret image but also incorporates a receiver authenticable mechanism for high security.

# B. Security Analysis

A detailed theoretical analysis of the security properties inherent in the proposed framework is presented. The analysis demonstrates that the framework affords security guarantees from several perspectives.

### B.1. Cover Selection

Existing generative image steganography methods based on diffusion models are confined to embedding secret information within a fixed latent space, *i.e.*, a fixed single time step of the diffusion process. This fixed-point conceal strategy grants any adversary a deterministic target for extraction attempts, thereby imposing significant security risks on the concealed secret data.

Departing from existing methods, the proposed steganography framework employs an adaptive strategy to dynamically and adaptively selects the optimal latent space (diffusion time step) for each secret image, rather than operating within a fixed embedding space (i.e., a predetermined diffusion time step). This results in a secret-dependent, flexible conceal location. Consequently, without prior knowledge of this location, an adversary must exhaustively search across all potential hiding spaces $\mathcal{Z} = \{z_1, z_2, \cdots, z_T\}$. For a diffusion model with $T$ time steps, this necessity reduces the adversary's success probability to $1/T$, a factor of T lower than that of methods employing a fixed embedding space. For instance, with $T = 50$, the probability drops to $0.02$.

To further enhance security, the proposed framework decomposes each secret image into $C$ fragments, embedding each fragment sequentially. Crucially, each fragment is assigned a distinct, optimally selected embedding space, expanding the effective search space to $C$ distinct subspaces per image. Consequently, to reconstruct the secret image, an adversary must not only search across all $T^C$ potential spaces $\mathcal{Z} = \{\{z_t^{(i)}\}\}, t = 0, 1, \cdots, T, i = 0, 1, \cdots, C$, but also correctly combine fragments across these subspaces. Only one specific combination corresponds to the original image, reducing the adversary's success probability to $1/T^C$. For example, with $T = 50$ and $C = 4$, this probability becomes $1.6 \times 10^{-7}$, making the successful extraction of secret data nearly impossible. This achieves an exponential expansion of the adversary's required search space, thereby providing a foundational security enhancement.

### B.2. Receiver Authentication

Existing methods lack a receiver authentication mechanism, enabling any party, including adversarial third parties, to extract the concealed secret data. To address this, a **Signature Authentication Controller** is incorporated within the proposed framework. Specifically, the sender signs the hiding space containing the secret data during the hiding process with the digital signature $D_{sig} = DS(k_{sym})$, and the receiver subsequently authenticates this signature, where $k_{sym}$ is the shared secret between the sender and the receiver, and $D_{sig}$ is a digital signature derived with the shared key. This mechanism ensures that only an authorized receiver with the correct key and corresponding model can successfully verify ($\tilde{D}_{sig} = D_{sig}$) and extract the secret. Additionally, in multi-receiver scenarios involving multiple secret images, this controller enforces receiver isolation, preventing any single receiver from accessing all concealed secret data.

### B.3. Key

Furthermore, a secret key is employed to achieve targeted conceal of specific secret data during the conceal process. This key is shared exclusively between the sender and the receiver. The security of the concealed secret data is guaranteed as long as the key remains secure.

## C. Implementation Details

### C.1. Experimental Setting

**Benchmarks and Datasets.** The proposed model utilizes the pretrained Stable Diffusion V1.5 to generate the stego images. For a precise evaluation of the proposed method's performance, we conduct a detailed comparison with SOTA generative image steganography methods based on the Stable Diffusion V1.5, like CRoSS (Yu et al., 2024) and DiffStega (Yang et al., 2024). Following the DiffStega (Yang et al., 2024), the IPAdapter (Ye et al., 2023) is the image encoder in Guidance Injection. Our model is implemented with PyTorch and trained on the DIV2K (Agustsson & Timofte, 2017) training dataset. The evaluation is performed on the UniStega (Yang et al., 2024) dataset, DIV2K (Agustsson & Timofte, 2017) test dataset and Stega260 dataset (Yu et al., 2024) at a resolution of $256 \times 256$.

**Evaluation Metrics.** To assess the quality of secret/reveal pairs, Peak Signal-to-Noise Ratio (PSNR) and Structural Similarity Index (SSIM) (Wang et al., 2004) is utilized as performance metrics. In addition, the Naturalness Image Quality Evaluator (NIQE) (Mittal et al., 2012) evaluates the naturalness of stego images.

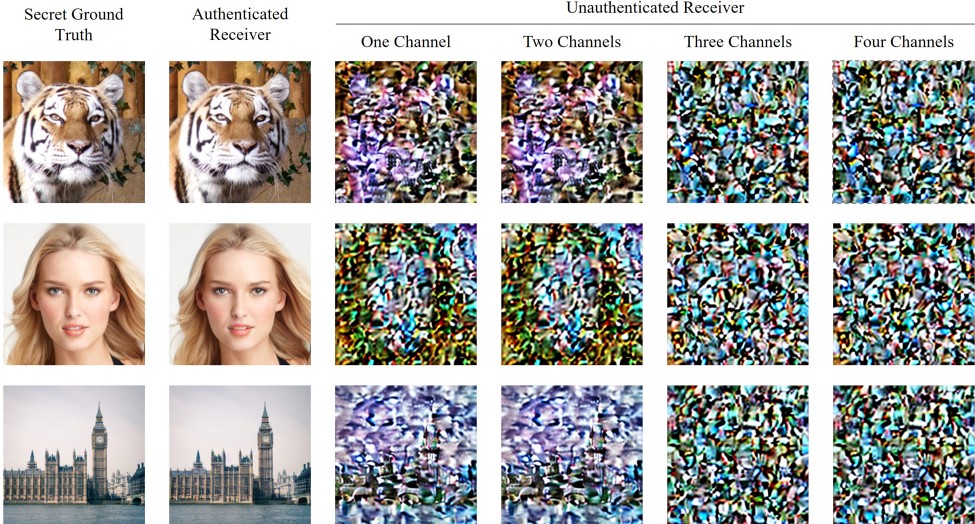

*Figure 8.* The effect of the Signature and Authentication Controller. The ground-truth secret images are shown in the first column, followed by the revealed secret image from authenticated receivers in the second. In contrast, columns 3 to 6 display the outputs for unauthenticated receivers, specifically when 1 to 4 channels are incorrectly localized.

*Table 7.* Quantitative results of revealed images on the UniStega dataset with degraded stego images.

| METHOD | CLEAN | GAUSSIAN BLUR | GAUSSIAN NOISE | JPEG COMPRESSION | POISSON |
|--------|-------|---------------|----------------|------------------|---------|
| ISN | 21.73 | 5.16 | 3.97 | 4.33 | 4.05 |
| HINET | 46.55 | 9.32 | 9.97 | 10.13 | 10.29 |
| CROSS | 21.25 | 20.08 | **18.97** | 20.20 | 19.27 |
| DIFFSTEGA | 23.29 | **20.85** | 18.62 | **21.16** | **20.15** |
| OURS | 26.59 | **21.84(0.99↑)** | **22.16(3.19↑)** | **22.94(1.78↑)** | **21.58(1.43↑)** |

# D. Additional Results

**Effect of the Signature and Authentication Controller.** The Signature and Authentication Controller is deployed to sign container during concealment and authenticate the receiver while revealing, thereby enforcing receiver isolation and enhancing the security of the concealed secret data. To validate this function, the reveal results of the proposed framework is evaluated for unauthenticated receivers, as detailed in Table 1. The data corresponds to an extraction attempt from a channel that failed authentication and a significant performance degradation is evident in the output for unauthenticated receivers. Specifically, lower similarity scores for recovery/secret image pairs unauthenticated receivers revealed, with a notable **8.42 dB** decrease in PSNR and **31%** reduction in SSIM, demonstrate the proposed model effectively bolsters security for recovery attempts with wrong keys. This demonstrates the controller's efficacy in ensuring receiver isolation and substantially improving the security of the concealed secret data.

Figure 8 presents contrasts the secret information extraction results for authenticated and unauthenticated receivers, presenting results for cases when authentication fails for 1 to 4 channels. The results clearly indicate that unauthenticated receivers are incapable of revealing the concealed secret data. The case of a single-channel failure is further detailed in Figure 4 and Figure 10. Collectively, these findings validate the efficacy of the proposed Signature and Authentication Controller in enforcing reliable receiver isolation and enhancing overall framework security.

**Effect of the Cover Optimization.** The Cover Optimization is introduced to mitigate the initial discrepancy and interference between secret data and the cover. This mechanism facilitates enriched secret feature extraction through structured redundancy augmentation, thereby improving reconstruction fidelity from 20.19dB to 23.41dB as evidenced in Table 5. These empirical results validate the encoding mechanism's dual functionality in dimensional adaptation and security enhancement, establishing its critical role in optimizing the proposed generative steganography systems.

**Robustness Analysis.** To evaluate the robustness of the proposed model, the proposed framework is tested under a variety of degradations, including Gaussian blur, Gaussian noise, JPEG compression, and Poisson noise. For Gaussian blur, a

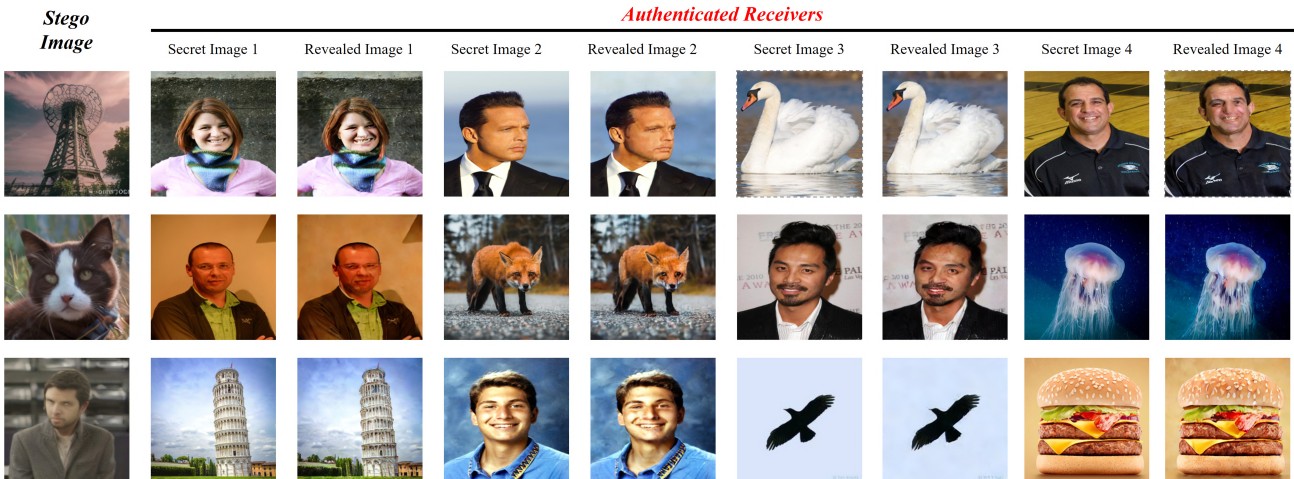

*Figure 9.* Multi-image steganography visual results of the proposed framework on the Stega260 dataset. The secret images are revealed with authenticated receivers.

Gaussian kernel with $\sigma = 0.1 \times std$ is applied to blur the stego images using convolution with a kernel size of 3, where $std$ denotes the standard deviation of the stego images. In the case of Gaussian noise, the noise is formulated as $\sigma = 0.1 \times std$. In addition, the quality factor is set as $Q = 75$ for JPEG compression in the experiments. The results, presented in Table 7 in terms of PSNR, indicate that the proposed model outperforms other SOTA models. These results demonstrate the favorable robustness of the proposed model against attacks.

**Complexity and Efficiency Metrics.** Computational efficiency was evaluated for the proposed and benchmark models using single-image steganography experiments on NVIDIA GeForce RTX 4090 and 3090 GPUs. As documented in Table 1, the proposed model demonstrated significantly reduced time overhead (5.29s/12.06s) and memory utilization (0.96GB/0.40GB) compared to DiffStega across both platforms. Relative to CRoSS, comparable efficiency was observed on the 4090 GPU, while the 3090 implementation achieved a 10.44s time consumption reduction with equivalent resource consumption. These empirical results demonstrate the model's deployability on Stable Diffusion V1.5-compatible architectures due to inherited computational characteristics.

**Visualization of Multi-Image Steganography.** Figure 9 visualizes the proposed framework's capability to conceal multiple secret images within a single stego image, with concealing four images with a single stego image for example. The results validate the framework's effectiveness in achieving large-capacity, multiple secret image generative steganography.

**Visualization of single Image Steganography.** Figure 10 illustrates the performance of the proposed model compared to other generative steganography models on the UniStega dataset, using three different prompts. When compared to CRoSS (Yu et al., 2024) and DiffStega (Yang et al., 2024), the proposed model significantly improves the naturalness and imperceptibility of the stego images. It also facilitates substantial modification of the secret image content, guided by the reference image, to minimize the similarity between the stego and secret image pairs. With style prompts, CRoSS (Yu et al., 2024) and DiffStega (Yang et al., 2024) also induce substantial deviation metrics between stego images and original secret images. However, under these operational constraints, CRoSS (Yu et al., 2024) and DiffStega (Yang et al., 2024) exhibit critical reconstruction fidelity degradation due to style-content entanglement. Furthermore, DiffStega (Yang et al., 2024) demonstrates critical information leakage vulnerabilities while unauthenticated receivers reveal the secret images. Conversely, the proposed model maintains robust security preservation against unauthorized receivers reveal with incorrect keys, as the exposed image differs drastically from the secret image and contains minimal secret information. Visual results indicate that the proposed model surpasses previous state-of-the-art (SOTA) models in terms of both effectiveness and security.

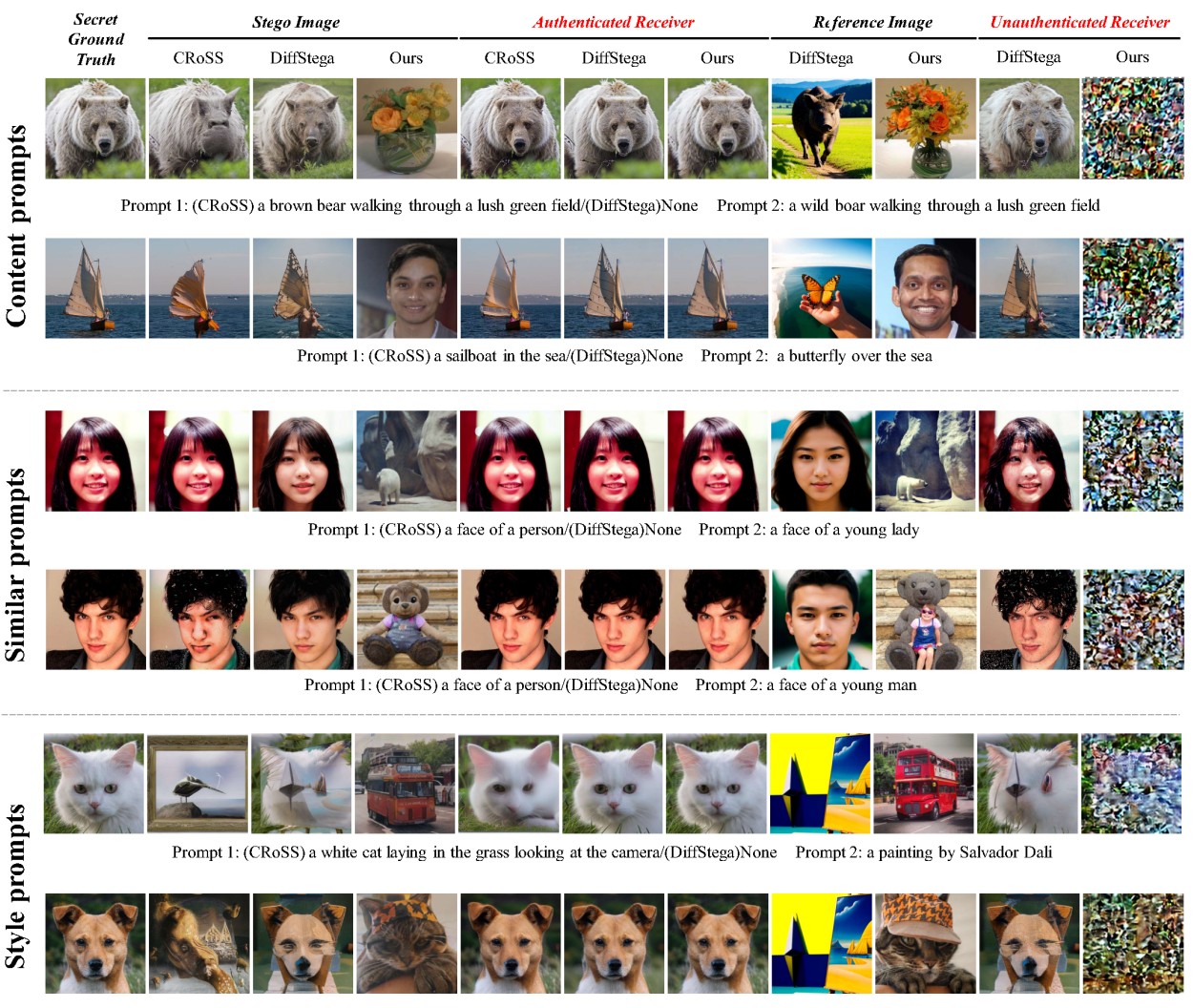

*Figure 10.* Single-secret steganography visualization results of our model versus state-of-the-art methods on UniStega, with the secret images are revealed with authenticated and unauthenticated receivers.

