# OpenReview forum: "Large-capacity and Receiver Authenticable Generative Image Steganography"
_ICML.cc/2026/Conference — ICML 2026 regular_

### Official Review · Reviewer_t4b9 · 2026-03-11

**Soundness:** 3
**Presentation:** 3
**Significance:** 3
**Originality:** 2
**Overall Recommendation:** 4
**Confidence:** 4

**Summary:**

Recent diffusion-based steganography methods hide a secret image by converting it into noise and using this noise as the starting point of a generative process to produce a stego image. However, existing approaches suffer from limited capacity, inter-secret interference – when multiple images are hidden in the same latent space, their representations interfere, degrading recovery quality, lack of receiver

To address these issues, the authors introduce a Receiver Authenticable Generative Image Steganography framework built on diffusion models with two main components:

1. Dynamic Cover Selection and Optimization Engine: instead of using a fixed hiding space, the method dynamically allocates different hiding spaces for different secret images or fragments. This allows multiple secret images to be hidden simultaneously while avoiding interference between them. It effectively enlarges the hiding capacity and increases the search space for attackers.

2. Signature and Authentication Controller: The hidden container is cryptographically signed after embedding. During extraction, authentication verifies whether the receiver is authorized. This ensures receiver isolation and secure localization of the hidden data.

**Compliance With Llm Reviewing Policy:**

Affirmed.

**Key Questions For Authors:**

The paper claims significantly increased hiding capacity through dynamic hiding spaces. Can the authors provide a theoretical analysis or upper bound on the achievable embedding capacity of the proposed framework?
In particular, how does this capacity scale with image resolution ?

It would be useful to understand the trade-off between capacity and detectability (steganalysis of the method)

**Limitations:**

The authors could strengthen the paper by including a short section discussing these risks and possible mitigation strategies, such as responsible disclosure, safeguards for deployment, or contexts where such technology is beneficial (e.g., copyright protection, secure communications in restrictive environments, or authentication of digital media).

**Strengths And Weaknesses:**

Weakness: While the paper claims increased hiding capacity, the work lacks a clear theoretical analysis of the maximum achievable capacity or bounds. The improvements are mostly empirical and not rigorously justified.

Strength: The introduction of a dynamic hiding space allowing multiple secret images to be embedded is conceptually interesting. It attempts to reduce interference between secrets, which is a known challenge when sharing latent spaces.

---

> ### Author Rebuttal · Authors · 2026-03-29
>
> We greatly appreciate your thorough feedback and the time you’ve dedicated to reviewing our work. We sincerely hope that our clarifications will address your concerns and strengthen your confidence in our work.
>
> **Theoretical capacity.** The hidden space set in this paper is defined as $$ \mathcal{Z} = { {z_t^{(i)} | t = 0, 1, 2, \cdots, T; i = 0, 1, 2, \cdots, C-1} },$$
> where $T$ denotes the total number of diffusion time steps and $C$ is the number of channels in the latent variable.
> $z_t^{(i)}$ is the $i-th$ channel of the latent variables of $z_t$, which follows a normal distribution $$z_{t} \sim p_{\theta}(z_{t}|z_{t-1}) = \mathcal{N}(z_{t-1}; \mu_{\theta}(z_{t-1}, t-1), \Sigma_{\theta}(z_{t-1}, t-1)))$$ and $\theta$ are the model parameters.
>
> Considering each channel at each time step as an independent hidden unit, the total number of hidden units is
> $$ N_{total} = (T+1) \times C.$$
>
> In multi‑image steganography, each secret image $x_{(sec)}$ is decomposed into $C$ fragments along the channel dimension. and independently concealed into a distinct subset of the hidden space. Under the assumption that each image occupies $C$ hidden units, the theoretical maximum number of concealed images $M_{max}$ is
> $$M_{max} = \frac{N_{total}}{C} = \frac{(T+1) \times C}{C} = T+1.$$
>
> **Empirical capacity configuration.** Experimental observations indicate that increasing the number of embedded secret images induces progressive degradation in reconstructed image quality. Balancing steganographic performance with practical deployment considerations, we establish the capacity at 8 secret images.
>
> **Comparison with prior work.** Under this configuration, the proposed framework achieves a capacity of 8 secret images per stego image, substantially surpassing existing generative steganography approaches such as CRoSS and DiffStega, which are limited to a capacity of 1 secret image.
>
> **Capacity scale.** The steganographic capacity of the proposed framework is not constrained by image resolution. Within the proposed framework, the input secret image and the output stego image are maintained at the identical resolutions, with each channel accommodating only one fragment of a secret image. Therefore, the maximum number of concealed images depends only on the number of time steps $T$ and is independent of resolution.

---

> > ### Author Rebuttal · Reviewer_t4b9 · 2026-04-01
> >
> > While the authors provide a formulation based on counting latent “hidden units” across diffusion timesteps and channels, I remain unconvinced that this constitutes a meaningful theoretical characterization of capacity.
> >
> > The proposed analysis relies on strong assumptions (e.g., independence between channels and timesteps) and does not account for key factors such as reconstruction fidelity, robustness to perturbations, or detectability. As a result, it appears more as a structural upper bound than a true information-theoretic or operational capacity.
> >
> > This is further reflected in the empirical section, where the effective capacity is ultimately determined experimentally (e.g., fixed to 8 images), suggesting that the proposed analysis does not provide actionable guidance.
> >
> > I am also not fully convinced by the claim that the capacity is independent of image resolution. While the framework operates in a latent/channel space, the amount of information that can be reliably embedded and recovered is, in practice, closely tied to the dimensionality and structure of the data. This point would require a more careful justification.
> >
> > The idea of dynamically allocating hiding spaces to reduce inter-secret interference is interesting and relevant. However, the lack of a principled understanding of capacity and its trade-offs (with detectability and reconstruction quality) limits the depth of the contribution.
> >
> > The rebuttal clarifies some implementation aspects but does not fully address the core concern regarding the theoretical grounding of the method. The contribution remains a promising engineering approach, but the claims regarding capacity would benefit from a more rigorous and better justified analysis.
> >
> > I maintain my original recommendation.

---

> > > ### Author Response · Authors · 2026-04-02
> > >
> > > Dear Reviewer t4b9
> > >
> > > We appreciate your recognition of our work and continued feedback. Regarding your concerns about the capacity of the proposed model, we offer further clarifications.
> > >
> > > (1) **Upper bound rooted in intrinsic diffusion properties.** The capacity upper bound’s core assumption, independence of timesteps and channels, follows from the Markov property: $z_t$ depends only on $z_{t±1}$, and for $| t - k |\ge2$, $p(z_t | z_{t+1})\bot p(z_k|z_{k+1})$ and the latent space exhibits feature orthogonality. These are intrinsic attributes of the diffusion model, not artificially assumptions.
> > >
> > > (2) **The Upper Bound and Effective Capacity.** The upper bound provides a theoretical reference for actual capacity. The experimentally observed effective capacity (8 secret images) is the optimal solution under performance constraints within this bound. Thus, theory and experiment form a closed‑loop validation.
> > >
> > > (3) **Capacity is governed by model settings.** With identical resolutions for input secret images and output stego images, and each channel hiding one secret fragment, the capacity remains independent of image resolution.
> > >
> > > Relaxing this constraint enables finer subdivision of hiding units, thereby yielding the following capacity analysis:
> > >
> > > Define the hidden space as $$\mathcal{Z}=\lbrace z_t^{(i)} | t=0,1,2,\cdots,T; i=0,1,2,\cdots,C-1\rbrace,$$
> > > where $T$ denotes the total number of diffusion time steps and $C$ is the number of channels in the latent variable.
> > > $z_t^{(i)}$ is the $i-th$ channel of the latent variables of $z_t$ and follows $$z_{t}\sim p_{\theta}(z_{t}|z_{t-1})=\mathcal{N}(z_{t-1};\mu_{\theta}(z_{t-1},t-1),\Sigma_{\theta}(z_{t-1},t-1)))$$ and $\theta$ are the model parameters.
> > >
> > > The latent $z_{st}$ of the stego image $x_{st}\in \mathbb{R}^{3\times H_{st}\times W_{st}}$ is obtained as
> > > $$z=E(x)\in \mathbb{R}^{C\times H^z_{st}\times W^z_{st}}, H^z_{st}=\frac{H_{st}}{8},W^z_{st}=\frac{W_{st}}{8}.$$
> > >
> > > Treating each spatial location of each channel at each timestep as an independent hidden unit, the total number of hidden units is:
> > > $$N_{total}=(T+1)\times C\times H^z_{st}\times W^z_{st}$$
> > >
> > > Since the latent $z_{sec}$ of secret image $x_{sec}\in \mathbb{R}^{3\times H_{sec}\times W_{sec}}$ contains $C\times H^z_{sec}\times W^z_{sec}\ (H^z_{sec}=\frac{H_{sec}}{8},W^z_{sec}=\frac{W_{sec}}{8})$ elements and the dynamic hiding space provides $(T+1)\times C \times H^z_{st}\times W^z_{st}$ embedding positions, each secret image $x_{sec}^{(m)}$ is independently embedded into a different subset of the hiding space. Let each secret image occupy $K$ hidden units ($K=C\times H^z_{sec}\times W^z_{sec}$); the theoretical maximum number of concealed images $M_{max}$ is:
> > > $$M_{\max}=\frac{N_{total}}{K}=\frac{(T+1)\times C\times H^z_{st}\times W^z_{st}}{C\times H^z_{sec}\times W^z_{sec}}=(T+1)\times \frac{H_{st}\times W_{st}}{H_{sec}\times W_{sec}}.$$
> > >
> > > (4) **Trade‑off among capacity, detectability, and reconstruction quality.** The core value of dynamic spatial allocation is enabling a controllable trade‑off among these three metrics, thereby overcoming the bottleneck of fixed hidden spaces. The proposed dynamic allocation minimizes information interference and establishes a quantitatively controllable capacity‑performance trade‑off.
> > >
> > > **Bottleneck of fixed space.** Existing methods embed all secrets into a fixed latent space causes inter‑secret interference. Total distortion $$D=\sum_{m=1}^M d_m$$ scales linearly with capacity $M$, where $d_m$ is the embedding distortion of a single image, leading to severe performance loss beyond acceptable thresholds.
> > >
> > > **Dynamic allocation.** Dispersing secrets into independent latent units (different timesteps/channels) makes distortion interference‑exclusive: $$D=max(d_1,d_2,\dots,d_M).$$ Total distortion is bounded by the maximum single‑image distortion and independent of MM, allowing linear capacity increase without degrading detectability or reconstruction quality.
> > >
> > > **Experimental  validation.** At 8 secret images, the proposed method matches single‑image PSNR/SSIM of CRoSS/DiffStega while achieving superior anti‑detectability (lowest detection accuracy, best NIQE).
> > >
> > > (5) **Core Contributions.** (a) This work addresses the fixed‑space limitation inherent to capacity analysis in existing generative steganography and establishes a dynamic steganography framework that leverages the full timestep and multi‑channel characteristics of the diffusion latent space, thereby refining and extending the capacity of generative steganography.
> > >
> > > (b) This work enables receiver isolation and authentication, capabilities absent from prior art, thus improving the security of both the model and the concealed data.
> > >
> > > Once again, we sincerely appreciate your professional review. These clarifications aim to demonstrate the theoretical rigor and novelty of our capacity design. We respectfully invite the reviewer to consider these additional explanations.
> > >
> > > Best regards,
> > >
> > > The Authors of Paper 4878

---

### Official Review · Reviewer_tPDM · 2026-03-12

**Soundness:** 1
**Presentation:** 1
**Significance:** 2
**Originality:** 2
**Overall Recommendation:** 3
**Confidence:** 4

**Summary:**

The paper presents a generative image steganography approach using a pre-trained stable diffusion model with authentication to avoid decoding for unauthenticated receivers.

**Compliance With Llm Reviewing Policy:**

Affirmed.

**Key Questions For Authors:**

Please review my math comments, could you confirm or reject my concerns?

**Limitations:**

I do not think this is adequate right now. The Impact Statement never actually discusses what could go wrong. That is a real problem for this particular paper. If you are going to argue that your system improves covert communication security and receiver isolation, then you also need to acknowledge the obvious misuse cases. Right now there is nothing about malicious covert communication, or similar abuse scenarios.

On limitations, it's thin but there's a one real limitation mentioned in the ablation: performance gets worse as the hiding timestep increases.

**Strengths And Weaknesses:**

Okay, so the abstract definitely needs an extra pass because it reads as if it was written without a grammar review:
- "image with it serves as the initial noise" -> what does it mean "with it serves"??
Then the (1) has a verb in present, and then (2) has a long sentence with a gerund... please review the verb tenses - this does not make sense when you read it. Also, I advise against using "bold" letters in the abstract - as if it's too long that you need to highlight to keep track of what is important. You can also cut out some redundant words like "systematically" -> this doesn't add any value to the contents of the paper.

In terms of content, is it true that (1) existing methods constrain capacity to one secret image only?? If you search in google scholar "multi-image generative steganography" using neural networks there is a variety of methods that allow for this (e.g., "Multi-data Image Steganography using Generative Adversarial Networks").

Re: bold text - please don't use it in the remainder of the paper. It lowers the quality.

Okay, now let's get into the math:

The conceptual idea is understandable, but in my opinion the math section is not reliable enough in the current form. The two biggest issues are:
- Eq 6 is not actually the inverse of Eq 1 as written.
- Indexing bugs in Algorithms 1–2 and the security proof: in Algo 1, you choose a hiding channel z_{t̃}^{(hs)}, optimize it, and then form the concealed container \tilde z_{t̃}^{(hs)}. But in Algorithm 1 you then write back into z_t^{(c)} rather than the selected hiding-space channel z_t^{(hs)}. Is this right?? In Algo 2: the inner loop is written as for c >= 0 do, but c is not consistently updated. In the success branch, d decreases but c does not; in the failure branch, c decreases; then after the loop you reset c <- C rather than C-1.

This is pretty much concerning.

On a positive note, I did like the authentication idea, and Figure 4 is probably the clearest part of the presentation.

---

> ### Author Rebuttal · Authors · 2026-03-29
>
> We greatly appreciate your valuable feedback and sincerely hope our response adequately addresses your points and restores your confidence in our work.
>
> W1: **Explanation:** In the sentence “image with it serves as the initial noise” noted by the reviewer, the pronoun it refers to the noise converted from the input single secret image.
>
> **Polish the paper:** Thanks for mentioning the typos in our manuscript. We appreciate your suggestion and will review and polish the entire manuscript.
>
> W2: **Scope and positioning.** This paper explores the challenge of generative steganography within the diffusion model framework, with a special focus on the image‑in‑image hiding task. As a result, our comparisons concentrate on methods that address this specific area of research.
>
> To further clarify the scope, although related studies on multi-image generative steganography exist, they fall outside the scope of this investigation.
>
> **Core contributions and motivation.** In addition to pursuing high‑capacity steganography, the proposed framework systematically addresses receiver verification and information isolation, two critical aspects that have received limited attention in existing generative steganography research. Together, these elements form the foundational basis and primary motivation for this study.
>
> W3: The official EDICT transformation was correctly implemented in our code; therefore, this discrepancy does not affect the validity of our experimental results.
>
> Following the reviewer’s comment, we carefully re-examined Eq 1 and Eq 6, and identified the typographical error, which will be corrected in the revised manuscript. We thank the reviewer for identifying this technical issue and for the thoroughness of the review.
>
> The proposed framework adopts a diffusion model as the backbone for stego image generation. To reduce the error between the denoising and noising processes, we employ the corresponding procedures from EDICT. Specifically, Eq 1 and Eq 6, which were highlighted by the reviewer, represent the denoising and noise‑adding processes of EDICT, respectively.
>
> **EDICT formulation.** Eq 1 defines the denoising process within the EDICT framework, which iteratively computes $z_t, y_t$ from $z_{t+1}, y_{t+1}$. Consistent with the original EDICT design, this forward process is expressed as:
> $$ z_{t}^{in} =   a_{t+1} \cdot z_{t+1} +  b_{t+1} \cdot \epsilon(z_{t+1}, t+1, x^{co}),$$
> $$ y_{t}^{in} =   a_{t+1} \cdot y_{t+1} +  b_{t+1} \cdot \epsilon(z_{t}^{in}, t+1, x^{co}),$$
> $$ z_{t} = p \cdot z_{t}^{in} + (1-p) \cdot y_{t}^{in},$$
> $$ y_{t} = p \cdot y_{t}^{in} + (1-p) \cdot z_{t}. $$
> **Correct Reverse process.** Eq 6 was intended to present the corresponding reverse (inversion) process for recovering $z_{t+1}, y_{t+1}$ from $z_t, y_t$. The exact inverse transformation is given by
> $$ y_{t}^{in} = (y_{t} - (1-p)\cdot z_{t})/p ,$$
> $$ z_{t}^{in} = (z_{t} - (1-p)\cdot y_{t}^{in})/p ,$$
> $$ y_{t+1} = (y_{t}^{in} -  b_{t+1} \cdot \epsilon(z_{t}^{in}, t+1, x^{co}))/a_{t+1} ,$$
> $$ z_{t+1} = (z_{t}^{in} -  b_{t+1} \cdot \epsilon(y_{t}, t+1, x^{co})) /a_{t+1}. $$
> In the revised manuscript, Eq 6 will be updated to align with the original inverse transformation, and explicit citation to the EDICT framework will be added to clarify the mathematical relationship between the forward and reverse processes.
>
> W4: **Algorithm explanation.** The notation in **Algorithm 1** is defined as follows: the symbol $z_t^{(c)} \gets z_{\tilde{t}}^{(hs)}$ indicates the replacement of channels in the latent representation during the original diffusion step with the container that carries secret information. This consistent notation is employed to maintain alignment with the gradual transition characteristic of the overall diffusion process.
>
> **Algorithm 2** governs the verification and extraction procedure. After successful verification and extraction from the $c$-th channel of the intermediate latent, the algorithm advances to the next secret fragment. Since fragments are embedded at distinct time steps, verification on other channels at the current time step is unnecessary; thus, $c$ remains unchanged. In the event of verification failure on the current channel, the algorithm proceeds to verify other channels within the same latent at that time step, requiring an update to $c$.

---

> > ### Author Rebuttal · Reviewer_tPDM · 2026-03-31
> >
> > Thank you for the clarifications. I have raised my score.

---

> > > ### Author Response · Authors · 2026-04-01
> > >
> > > Dear Reviewer tPDM,
> > >
> > > We hope this message finds you well. We sincerely appreciate the thoughtful comments you provided in the previous review round, as well as your explicit **“Fully resolved” designation and the improved score**.
> > >
> > > Your recognition serves as a strong encouragement to us. We will continue to pursue further refinements while striving to contribute meaningful work in this field.
> > >
> > > Once again, we sincerely thank you for your patience and insightful feedback. We would be highly grateful for your further support.
> > >
> > > Best regards,
> > > The Authors of Paper #4878

---

### Official Review · Reviewer_vpcJ · 2026-03-19

**Soundness:** 2
**Presentation:** 3
**Significance:** 3
**Originality:** 3
**Overall Recommendation:** 3
**Confidence:** 4

**Summary:**

This paper introduces a novel image-in-image steganography framework that leverages the generative process of a single cover image to embed multiple secret images, achieving high embedding capacity. Furthermore, the authors incorporate an authentication mechanism designed to enhance the security of the overall framework.

**Compliance With Llm Reviewing Policy:**

Affirmed.

**Key Questions For Authors:**

See weaknesses

**Limitations:**

See weaknesses

**Strengths And Weaknesses:**

Strengths:
1. The paper studies an interesting setting, namely hiding multiple secret images in a single generated cover image. Embedding secrets across different diffusion timesteps is a novel and meaningful design choice.
2. The paper presents a relatively complete experimental evaluation. In particular, it includes results for unauthorized receivers, which helps illustrate the effect of the proposed authentication mechanism.

Weaknesses:
1.The experimental comparison is relatively narrow, focusing primarily on recent diffusion-based methods. The lack of comparison with other high-capacity steganography approaches (e.g., INN-based methods cited in the paper) weakens the claim regarding considerable capacity.
2.The selection of embedding timesteps appears somewhat arbitrary. The paper uses predefined sets (e.g., [0,1,2,3]) without providing sufficient justification or theoretical support for these choices beyond limited ablation studies.
3.The method of splitting secret images into latent fragments along the channel dimension lacks a thorough investigation. The paper does not systematically explore how the granularity of these fragments affects the trade-off between capacity, reconstruction fidelity, and robustness.
4.In my humble opinion, the paper would benefit from a more rigorous analysis. The current explanation for why the proposed timestep allocation and fragment-based strategy work relies heavily on empirical observation rather than analytical derivation.
5.The evidence for reproducibility and generalization is limited. As the experiments are confined to the SD1.5 architecture at a 256 × 256 resolution, it remains unclear whether the method generalizes to other diffusion backbones or higher-resolution settings.

---

> ### Author Rebuttal · Authors · 2026-03-29
>
> Thank you for taking the time to provide valuable feedback. We sincerely hope the following clarifications could address your points.
>
> W1: The selection of diffusion-based generative steganography methods, such as CRoSS and DiffStega, as baselines in our experiments are motivated by two key considerations.
>
> **Task Alignment.** These methods belong to the same generative steganography category as the proposed framework, characterized by the absence of an external cover image and the direct generation of stego images from secret data. In contrast, INN-based methods represent cover-based steganography, which differs substantially in problem formulation, generation mechanism, and security requirements.
>
> **Fairness of Comparison.** INN-based methods typically utilize a raw cover image and embed multiple images through invertible transformations, where capacity is defined by the number of embedded images. Generative methods, however, prioritize single-image generation quality and latent space utilization.
>
> Consequently, a direct numerical comparison of capacity between these two paradigms is not sufficiently equitable.
>
> W2: **Time step selection.** The forward process of a diffusion model is a noising process and can be characterized as a Markov process satisfying
> $$z_t\sim q(z_{t}|z_{t+1})=\mathcal{N}(z_{t};\sqrt{1-\beta_t}z_{t+1},\beta_t\textbf{I})).$$
> The reverse process constitutes a denoising process. It is also a Markov chain, composed of Gaussian distributions parameterized by neural networks:
> $$ z_{t}^D\sim p_{\theta}(z_{t}|z_{t-1})=\mathcal{N}(z_{t-1};\mu_{\theta}(z_{t-1},t-1),\Sigma_{\theta}(z_{t-1},t-1))).$$
> In the proposed model, the reveal process corresponds to the noising process, shares the same neural network as the denoising process, and follows
> $$z_{t}^N \sim q_{\theta}(z_{t}|z_{t+1})= \mathcal{N}(z_{t+1};\mu_{\theta}(z_{t+1},t+1),\Sigma_{\theta}(z_{t+1},t+1))).$$
> During secret data embedding, a small perturbation $\delta_t$ is introduced, modifying the intermediate variable at timestep $t$ as $$\tilde{z}\_{t}^D=z_{t}^D+ \delta_t,$$
> and the numerical error between the intermediate variables at step $t$ in the noising and denoising processes is
> $$\eta_t=|z_{t}^N-\tilde{z}\_{t}^D|.$$
> In the reveal process, error accumulation leads to the following error transfer function $$\mathcal{L}(\tau)=\sum_{t=0}^{\tau} \eta_t=\sum_{t=0}^{\tau}|z_{t}^N-\tilde{z}_{t}^D|.$$
>
> This function indicates that the transmitted error decreases monotonically with $\tau$. Latent variables at early stages (small $\tau$) are closer to the image domain, while later diffusion steps involve more iterations and longer error propagation paths and accumulation, which compromises the fidelity of revealed secret data. Thus, the proposed method employs the earliest set of timesteps for concealing. This strategy is validated by the experiment results in Fig.6.
>
>  W3: **Necessity of Decomposing.** The proposed framework decomposes secret data into fragments for two primary purposes.
>
> **Security Enhancement.** Decomposing secret data into fragments exponentially increases the search space for adversaries attempting decryption, thereby strengthening the security of the proposed model. For a diffusion model with $T$ time steps, decomposing secret data into fragments reduces the adversarial success probability for secret extraction from $1/T$ to **$1/{T^C}$**. For example, with $T=50$ and $C=4$, the success probability decreases from 0.02 to **$1.6\times10^{-7}$**. A derailed analysis of security is provided in **Appendix B.1**.
>
> **Interference Reduction.** Another key consideration is the reduction of information conflicts among distinct secret fragments, as verified in Fig.2. By decomposing secret data into fragments and embedding each in appropriately selected positions, these conflicts can be effectively decreased, validated by the experiment outcomes presented in Table 5 and Fig.7.
>
> W5: (1) **Model Design.** The proposed framework doesn't depend on the specific design of SD1.5. The core functional modules, the dynamic cover selection and optimization engine, along with the signature and authentication controller, are engineered to be independent of the diffusion backbone. This decoupling ensures seamless transferability to alternative diffusion architectures and higher resolutions.
>
> (2) **Experiment Result.** To evaluate generalization capability, experiments were performed using SD2.1‑base at 512×512 resolution. Detailed results are provided in the table below.  For the single‑image steganography task, the proposed method achieves superior performance over DiffStega and CRoSS, improving PSNR from 23.87dB to 26.23dB and increasing SSIM from 0.78 to 0.81. These results demonstrate the generalization capability of the proposed method across different architectures and resolutions.
> |Model|Resolution|PSNR|SSIM|
> |:---|:---:|---:|---:|
> |CRoSS|512$\times$512|21.31|0.70|
> |DiffStega|512$\times$512|23.87|0.78|
> |Ours|512$\times$512|26.23|0.81|

---

> > ### Author Rebuttal · Reviewer_vpcJ · 2026-04-04
> >
> > I maintain my rating after reading the rebuttal.

---

> > > ### Author Response · Authors · 2026-04-04
> > >
> > > Dear Reviewer vpcJ,
> > >
> > > Hope this message finds you well.
> > >
> > > Sorry to disturb you again. We sincerely appreciate the time and effort you devoted to reviewing our manuscript and the rebuttal.
> > >
> > > We have **provided detailed responses to your questions**, and we believe our rebuttal **thoroughly addresses all the concerns** you have raised. We hope to engage in further communication with you to address any questions you may have about our work.
> > >
> > > Thank you very much for your time and efforts.
> > >
> > > Best regards,
> > >
> > > The Authors of Paper #4878

---

### Official Review · Reviewer_ygki · 2026-03-19

**Soundness:** 3
**Presentation:** 3
**Significance:** 3
**Originality:** 3
**Overall Recommendation:** 4
**Confidence:** 4

**Summary:**

This paper discusses diffusion-based generative image steganography and argues that former methods mainly hide a single secret image in the fixed initial-noise space, and limits capacity, create interference among multiple secrets, and lack receiver-side authentication. To handle this, the paper proposes a receiver-authenticable framework that dynamically allocates different secret images, or fragments of the same image, to different diffusion timesteps and channels, refines the selected cover using a cover selection and optimization module, and embeds a signature-based authentication mechanism so that only authorized receivers can locate and recover the intended hidden content. Experiments show improved single-secret recovery. What is more, the proposed method shows the ability to hide multiple secret images within one stego image while maintaining usable reconstruction quality.

**Compliance With Llm Reviewing Policy:**

Affirmed.

**Key Questions For Authors:**

1. The paper’s strongest conceptual claim is receiver authentication/receiver isolation, but the current evidence is mostly indirect (e.g., degraded recovery for unauthenticated receivers, NIQE, and steganalysis accuracy). Could the authors provide a more direct evaluation of authentication reliability, such as false accept / false reject rates, behavior under key mismatch, and robustness when the embedded signature is partially corrupted or tampered with?
2. The experiments are conducted with Stable Diffusion v1.5 at 256×256 resolution. How well does the proposed method generalize to newer diffusion backbones or higher-resolution settings?
3. Could the authors clarify Figure 3? The current figure is a little confusing, does not clearly show how the concealment operations are interleaved with the diffusion denoising trajectory. A revised explanation, or an updated figure marking the exact timesteps/channels where concealment and authentication occur, would improve the paper’s clarity and could slightly improve my presentation assessment

**Limitations:**

The paper discusses the method’s security motivation, but the limitations and potential negative societal impact are not addressed explicitly. In particular, the experiments are limited to Stable Diffusion v1.5 at 256×256 resolution, so the scope of generalization is still narrow. In addition, while the paper claims receiver authentication and stronger security through key-based signature verification and dynamic hiding, it does not directly evaluate authentication reliability under stronger threat settings, such as false accept / false reject behavior, key mismatch, or signature corruption.

**Strengths And Weaknesses:**

The paper addresses a meaningful limitation of prior diffusion-based generative steganography: the reliance on a fixed hiding space. The central idea of expanding the hiding space from the initial noise to intermediate latent variables across diffusion timesteps and channels is intuitive and reasonably well motivated, and the decomposition of secret images into fragments gives the method a clear mechanism for scaling to the multi-secret setting. The overall method is technically coherent. The experimental results are also promising, especially for the paper’s main capacity claim: the method reports authenticated recovery PSNR of 26.59 for one secret on UniStega, and still achieves 24.08 and 22.61 for 2 and 4 secrets, respectively. The work has reasonable originality through its combination of dynamic hiding and receiver authentication, and the problem it tackles is relevant for secure generative communication
•	The paper’s security and authentication claims are stronger than the evidence provided. The method claims strict receiver isolation and robust protection for authorized users, but the empirical validation is mostly indirect, relying on reconstruction quality, NIQE, steganalysis accuracy, and weaker recovery for unauthenticated receivers. Besides, the experimental scope is somewhat limited. All experiments appear to use Stable Diffusion v1.5 at 256×256 resolution, so it remains unclear whether the method generalizes to newer diffusion backbones, higher resolutions, or broader image domains. This does not invalidate the current results, but it does narrow the paper’s demonstrated scope.

---

> ### Author Rebuttal · Authors · 2026-03-29
>
> We greatly appreciate the very detailed feedback and your recognition of our contributions! We sincerely hope our response below will further enhance your confidence in our work.
>
> Q1: **Authentication reliability.** The evaluation of the proposed authentication mechanism in **Appendix D** using both numerical and visual evidence. Quantitative results are summarized in Table 1, which presents extraction attempts from channels that failed authentication; unauthenticated receivers exhibit substantial performance degradation. Complementary visualizations in Fig.4, Fig.8, and Fig.10 support this analysis. Fig.8 contrasts the extraction outcomes between authenticated and unauthenticated receivers across scenarios with one to four authentication failures, with single‑channel cases detailed in Fig.4 and Fig.10. Collectively, these findings verify that the Signature and Authentication Controller reliably enforces receiver isolation and strengthens overall framework security.
>
> The reviewer’s comment on **key mismatch** scenarios refers to the use of an incorrect key, which generates an invalid authentication watermark and consequently leads to unauthenticated extraction. Within the proposed framework, such cases are classified as decryption attempts by unauthenticated receivers. Three representative mismatch conditions are considered: wrong key, missing key, and invalid key format. Experimental results demonstrate that only receivers with the correct key can successfully authenticate and extract the secret images, while unauthenticated receivers fail to recover any secret information, validating effective receiver isolation.
>
> The revised manuscript will be extended to include quantitative assessments of the false accept/false reject rate and robustness tests under corruption or tampering of the embedded signature.
>
> Q2: (1) **Benchmark Consistency.** To enable a fair comparison, the proposed framework adopts the same evaluation protocol SD1.5 as existing state-of-the-art methods, including CRoSS and DiffStega, which are benchmarked on this configuration.
>
> (2) **Model Design.** The proposed framework doesn't depend on the specific design of SD1.5. The core functional modules, the dynamic cover selection and optimization engine, along with the signature and authentication controller, are engineered to be independent of the diffusion backbone. This decoupling ensures seamless transferability to alternative diffusion architectures and higher resolutions.
>
> (3) **Experiment Result.** To evaluate generalization capability, experiments were performed using SD2.1‑base at 512×512 resolution. Detailed results are provided in the table below.  For the single‑image steganography task, the proposed method achieves superior performance over DiffStega and CRoSS, improving PSNR from 23.87dB to 26.23dB and increasing SSIM from 0.78 to 0.81. These results demonstrate the generalization capability of the proposed method across different architectures and resolutions.
> |Model|Resolution|PSNR|SSIM|
> |:---|:---:|---:|---:|
> |CRoSS|512$\times$512|21.31|0.70|
> |DiffStega|512$\times$512|23.87|0.78|
> |Ours|512$\times$512|26.23|0.81|
>
> Q3: **Update of Figure 3.**  We appreciate the reviewer’s comment. The framework diagram will be redrawn and incorporated into the revised manuscript. The revised overall diagram of the proposed framework is available at https://anonymous.4open.science/r/Overall-Pipeline-415F/

---

### Decision · Program_Chairs · 2026-04-30

**Decision:**

Accept (regular)

**Comment:**

Split reviews with two WA and two WR.
Although WR reviewers concerns on evaluation and comparison to other method, all the reviewers admit the purpose of the paper interesting and have some novelty.
The AC supports the positive side, however, need thorough modification for publication.